# Obesity and Related Type 2 Diabetes: A Failure of the Autonomic Nervous System Controlling Gastrointestinal Function?

**Claudio Blasi**

Centro Diabete, Ospedale Sandro Pertini-ASLRMB, Largo T. Solera n.10, 00199 Rome, Italy; diabcb@gmail.com

**Abstract:** The pandemic spread of obesity and type 2 diabetes is a serious health problem that cannot be contained with common therapies. At present, the most effective therapeutic tool is metabolic surgery, which substantially modifies the gastrointestinal anatomical structure. This review reflects the state of the art research in obesity and type 2 diabetes, describing the probable reason for their spread, how the various brain sectors are involved (with particular emphasis on the role of the vagal system controlling different digestive functions), and the possible mechanisms for the effectiveness of bariatric surgery. According to the writer's interpretation, the identification of drugs that can modulate the activity of some receptor subunits of the vagal neurons and energy-controlling structures of the central nervous system (CNS), and/or specific physical treatment of cortical areas, could reproduce, non-surgically, the positive effects of metabolic surgery.

**Keywords:** GI role in energy homeostasis; obesity pathogenesis; obesity therapy; metabolic surgery mechanism; vagal control of GI function; NMDA receptors; pharmacotherapy of obesity

## 1. Introduction

Diabetes and obesity have been classified as the third and fourth leading health risk factors, respectively, in the world [1]. Their growing prevalence cannot be countered or even limited, and they are considered to be epidemics. Adiposity-based chronic disease is probably the greatest noninfectious epidemic of the 21st century; for example, half of all Americans are expected to be affected by obesity in about ten years [2].

*Relationship between Obesity and Diabetes*

There is a close relationship between obesity and type 2 diabetes mellitus, to the extent that the term diabesity has been coined. In fact, diabetes is accompanied by obesity (and overweight) in 90% of cases [3], and is probably primarily responsible for the morbidity and mortality of obesity. The precise mechanisms of this close connection are still largely unexplored. However, some phases of the pathophysiological processes can be identified that have as their basis resistance to insulin. This resistance occurs in patients with excess abdominal adipose tissue and raised amounts of circulating free fatty acids (FFA) [4] in the presence of steatotic (NAFLD) and insulin-resistant liver [5]. FFAs determine the blocking of post-receptor mechanisms of the glucose metabolism (glycogen synthesis) in target tissues, mainly muscle and adipose tissue. Initially, beta cells increase insulin secretion, but, over time, compensation becomes insufficient, gradually leading to hyperglycemia [4]. It is well-known that hyperglycemia involves a series of harmful consequences affecting, in particular, the cardio-circulatory system (endothelial damage) and the peripheral nervous system.

## 2. Why Is the Spread of Obesity So Rapid?

### 2.1. Lifestyle Change

The cause of the rapid spread of obesity is partly related to the spread of a lifestyle characterized by a marked reduction in physical activity and a high-calorie diet. Humans have a spontaneous attraction to high-calorie foods [6,7]. In fact, the human brain is structured according to the choice of palatable high-calorie foods, which was an evolutionary adaptation to cope with the uncertain food availability of our ancestors [6,7]. In the present situation, in which high-calorie food is abundantly available, this predisposition has become "maladaptive"—that is, more harmful than useful [8,9]. The motivation for eating has become predominantly hedonic (dependent on the search for pleasure), and is based in the hypothalamus, compared to homeostatic motivation (dependent on the energy need), which is based in the dopaminergic system [7].

### 2.2. Role of Epigenetics: Development Programming

However, changing lifestyles are insufficient to justify the rapid expansion of obesity. The main cause is to be found in genetic adaptations [10], which cannot involve a change in the DNA sequence (because this would require a very long time to manifest). For example, human twins demonstrate a phenotypic divergence in relation to obesity that occurs in just one generation, too quickly to be explained by a basic mutation rate [10,11]. Rather, the change involves epigenetic modifications, a hereditary process independent of the classical Mendelian laws [12].

Epigenetics can be defined as molecular mechanisms that establish and maintain mitotically stable variations of gene expression involving DNA methylation and histone methylation without altering the genetic sequence [13,14]. These epigenetic marks are established in the parent's germ line (sperm or egg cells) (genetic imprinting) and are maintained through mitotic cell divisions in the somatic cells of an organism [15]. During specific periods (e.g., pre-conception, particularly the period immediately after oocyte fertilization, gestation, and the first years of life), tissues and organs are particularly sensitive to numerous environmental stimuli and lifestyle factors (such as type of diet and physical activity) that affect susceptibility to disease during life [13].

Unlike the slow evolutionary processes of natural selection, epigenetics causes organisms to adapt during the first phase of fetal development in a manner that renders them able to cope with changes in the environment [16]. In this phase, called development plasticity, epigenetic modifications are known to be of crucial importance in the development of neural circuits responsible for energy homeostasis and in the integration of autonomic reflexes [17,18].

### 2.3. How Can Negative Environmental Conditions Influence The Development of Obesity?

In animal models, maternal obesity and consumption of high-calorie diets (rich in saturated fats and sugars) at the time of conception, during gestation, and during breastfeeding, have been shown to alter the normal development of the hedonic brain system in offspring that includes reward-related stimuli [16,19]. According to the theory of development programming [20], maternal conditions cause adaptive epigenetic changes in the fetus that influence the development of its organs, particularly the brain [16], in view of an adverse post-natal environment [21]. Nutritional excesses are among the most important stimuli of negative fetal programming [13]. When the pre-natal and post-natal environments do not match (e.g., pre-natal undernutrition followed by abundance of post-natal food), the risk of metabolic diseases increases, as widely confirmed in animals [13]. Furthermore, stress in the early stages of development can alter the serotonergic system, which plays an important role in controlling energy homeostasis. In the case of obesity, a metabolic aptitude to caloric accumulation is transferred to the unborn child who, in a subsequent environment with abundant food availability, would be predisposed to the obesogenic phenotype (evolutionary mismatch) [16]. This phenotype is transmitted to subsequent generations even in the absence of further environmental stimuli and, indeed, despite their absence (transgenerational effects) [22] (Figure 1).

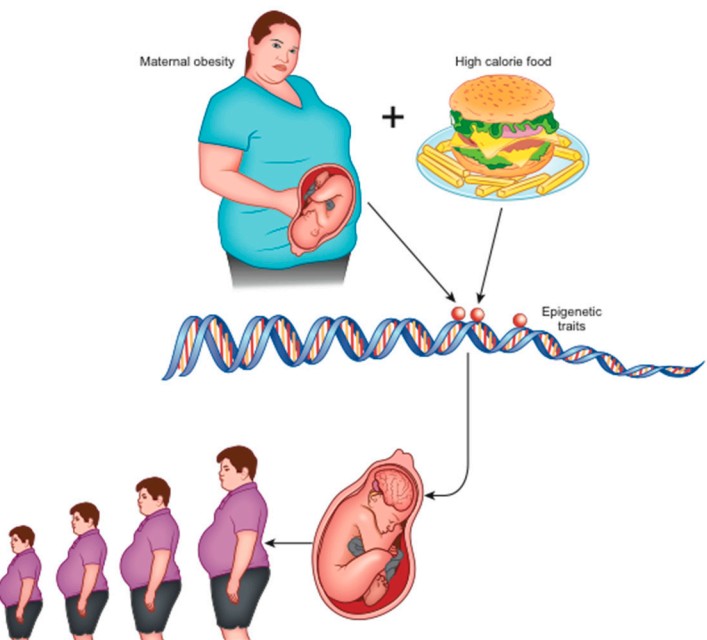

**Figure 1.** Maternal obesity and overeating at the origin of epigenetic changes that lead to neurologic-based generational predisposition to obesity.

Early and long-term exposure to a high-calorie diet during the intra-uterine and post-natal period causes alterations in the dopaminergic circuits that regulate the reward [16]. In addition, perinatal exposure to a high fat diet (HFD) alters the gastrointestinal vagal neuro-circuits in adulthood, which can result in predisposition to the development of obesity and related diseases through visceral effects [23]. This occurs because the evolution of the GABAA receptor present on dorsal motor nucleus of the vagus (DMV) neurons is blocked, maintaining activity in its subunits n.2/3 [23]. This increases the sensitivity to GABA-mediated inhibition by the nucleus tractus solitarius (NTS), and the consequent increase in the GABAergic tonic inhibition of DMV neurons involves a decrease in efferent output of the vagal neuron directed towards abdominal organs (such as the liver and endocrine pancreas) with digestive and metabolic consequences [24]. This flaw is maintained in adulthood [23].

In animal models, it has also been shown that a change in synaptic activation (synaptic plasticity) is induced in the reward system [16] followed by an inadequate response to the consumption of highly energetic food in concomitance with the pregnancy period [10,25]. This is due to the epigenetic alteration of the expression of dopamine and opioid-related genes present in the mesocorticolimbic circuits [26].

In rodents, paternal exposure to a type of diet and lifestyle, and to obesity, has also been shown to be associated with an increased risk of obesity in descendants, over multiple generations, through modification of the sperm epigenome [13,27].

Thus, generational epigenetic modifications in the development of the brain structures that control energy homeostasis and food intake are most likely to be the main reason for the pandemic spread of obesity, and could explain why it is so difficult to contain it [7].

## 3. How Could the Dysfunction of Energy Homeostasis Control Be Responsible for Obesity?

The control of energy homeostasis is performed by various brain sectors that interact with each other [28]. It should be considered that the brain functions are an integrated whole and that in the various functional areas, only a predominant role can be identified [9] (Figure 2).

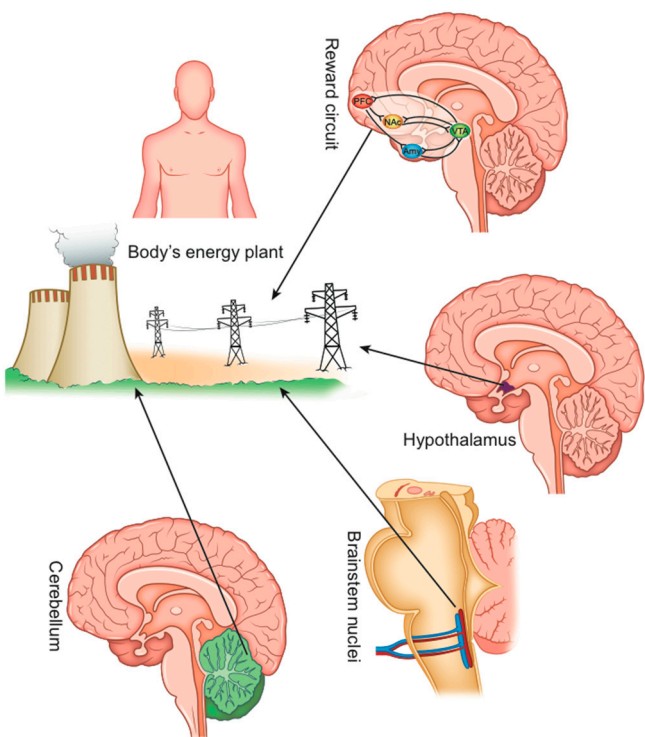

**Figure 2.** The main sectors of the CNS that control the body's energy homeostasis.

### 3.1. What Is the Role of the Reward System Dysfunction? Can We Take into Account a Food Addiction?

The emotional aspects (sensitivity to rewarding stimuli) of non-homeostatic eating behavior (i.e., hedonic food consumption, not activated by energy needs) are determined by the reward system, a grouping of dopaminergic brain structures and neuronal pathways (the ventral striatum, the amygdala, the anterior insula, the ventromedial prefrontal cortex, and the orbital-frontal cortex) that communicate with each other through the dopamine neurotransmitter and the opioids [9,29].

As with the abuse of drugs, appetizing foods are able to trigger the release of dopamine in the brain's nucleus accumbens. The nucleus accumbens (NAc or NAcc) (a region in the basal prosencephalon) that serves the cognitive processing of motivation, aversion, and gratification, and the ventral tegmental area (VTA), are central components [8,30,31]. The opioid system, in which the pleasure of eating originates, plays an important role (interacting with dopamine), with a wide secretion of opioid precursor proteins and the expression of their receptors [7,26]. Drugs of abuse and overfeeding modify this system to the extent that it supports compulsive seeking and taking drugs or food, with the loss of the ability to control strong impulses despite the awareness of its harmful consequences [31].

Equally, chronic consumption of rewarding foods can cause a downregulation of dopamine receptors to attenuate their over-stimulation [31]. Accordingly, striatal D2 receptor availability is lower in obese individuals and is negatively correlated with the body mass index (BMI) [32]. Decreased dopamine receptor availability may cause or exacerbate increased food intake as a form of self-medication in which the individual tries to compensate for a reduced reward experience [32].

Obesity is associated with changes in how the reward system reacts to food [33]. An elevated dopamine secretion induced by high-calorie foods causes an initial state of well-being and euphoria, which leads to eating even more [34]. The consequent intense and repeated secretion of dopamine causes a decrease in the sensitivity (downregulation) of its receptors (neuro-adaptation in the motivational circuit) [9]. This adaptation of the synaptic transmission (synaptic plasticity) progressively evolves into "maladaptation": to obtain the same response of the post-synaptic neuron, increasing neurotransmitters and, therefore, food (overfeeding) is needed until a deep

hyposensitivity is reached [8]. Some individuals may have a genetic predisposition to deficient dopamine signaling in the reward system, which causes them to increase DA levels in the brain with repetitive drug administration. This widely accepted theory is called "Reward Deficiency Syndrome" or "Hypodopaminergic Theory" [35]. By reducing food intake, withdrawal symptoms appear, such as anxiety, nervousness, dysphoria, and insomnia in both humans [36] and experimental animals [37], and the appetitive and consumer behavior is accentuated [9]. This helps to explain weight loss resistance and why any weight loss induced by calorie restriction recovers very quickly [9,27].

This obesity onset model also presupposes an imbalance between the reward circuits and the prefrontal cortical areas that wield cognitive control over food by inhibiting excessive behavior (the anterior cingulate cortex (ACC), the lateral prefrontal cortex (LPFC), and the left dorsolateral prefrontal cortex (DLPFC)) [38]. In obese patients, cortical inhibitory control is weaker [39]. This has also been correlated with the lower availability of dopamine receptors [32].

The food addiction mechanism (which consists of the transition from excess sporadic intake to compulsive intake with loss of control [40]) is very similar to that of recreational drug addiction or addiction from other causes [8,9,36,41]. The main common feature is the origin of the mesolimbic dopaminergic system [7,8]. There is evidence that some individuals are genetically vulnerable (through the epigenetic modifications described above) to a reduced ability to maintain the correct reward activity and cortical inhibition [31].

### 3.2. Role of the Brainstem and the Gut–Brain Axis

The vagal brainstem neurocircuits that regulate food intake and energy homeostasis, through the disorders of their synaptic connections, play an important role in the pathogenesis of obesity and type 2 diabetes mellitus [42].

A large number of endocrine cells (EC) are present on the wall of the digestive tract, scattered among the epithelial cells [43]. The EC are strategically placed close to the vagal afferent terminals, so that the first response to the various enterohormones is paracrine [42].

Activated by food, these produce more than 30 different neurotransmitters and hormonal peptides (including GLP-1, PYY, CCK, and GIP), which have a mainly local (paracrine) action. These enteric hormones stimulate the receptors of a vast network of vagal nerve fibers (called the intestinal brain or enteric nervous system) embedded in the lamina propria (the thin layer of connective tissue at the base of the mucosa that lines the gastrointestinal tract) and are composed of thousands of ganglia and about 500 million neurons (five times as many as present in the spinal cord), capable of operating autonomously and producing inputs towards the brain [42].

More than 90% of the body's 5-hydroxytryptamine (5-HT) and 50% of its dopamine are produced in the gastrointestinal tract, predominantly by the EC and, to a lesser extent, by myenteric neurons and mast cells that transmit their impulses to populations of neurons mainly in the nucleus tractus solitarius (NTS) of the brainstem [44].

Vagal afferent neurons (VAN), located in the nodosum ganglion (NG), innervate the intestine and terminate in the NTS. These afferent vagal fibers integrate gut-derived signals to regulate meal size [42]. From the NTS, impulses start towards other regions of the brain, including the hypothalamic and limbic nuclei, which play a significant role in the processing of neuroendocrine, behavioral, and autonomic functions related to weight control [45] and modulating their functions (such as motivation and pleasure): this is the so-called gut–brain axis [43]. In addition, through the adjacent dorsomedial nucleus (DMV), which contains the parasympathetic preganglion motor neurons that provide vagal motor output returning to the upper GI tract, the NTS connects with the main splanchnic organs that control the metabolism, i.e., the endocrine pancreas and the liver. In the condition of obesity and following a high-calorie diet, the sensitivity of the vagal afferent fibers that reach the NTS decreases and the activity of the efferent branch of vagus-vagal reflexes mediated by the DMV decreases [35]. A high-calorie diet alters vagally mediated satiety, resulting in overeating [46].

The deterioration of the synaptic connection with the DMV (synaptic maladaptation) leads to a disordered modulation of the splanchnic organs with their consequent dysfunction (altered production of insulin and glucagon, and impaired hepatic synthesis and release of glucose) [47].

The NTS also integrates inputs from the higher brain regions responsible for energy homeostasis and appetite control, and from circulating mediators, orchestrating a coherent output reflex to the splanchnic organs [48].

### 3.3. Importance of Impulses from the Oropharynx

Oro-sensory exposure (the release of nutrients, odors, and taste molecules from the food matrix into the mouth during oral processing) is mediated by chemosensory neurons in the oral cavity that project their afferent fibers to the NTS where the taste signals are processed and re-transmitted to the reward system [49].

The intermediate NTS receives fibers from the lingual-tonsillar branch of the IX cranial nerve, which carries additional gustatory and somato-sensory information [42]. More rostrally, the intermediate NTS also receives afferent fibers from the larynx and the lingual branch of the trigeminal nerve, from the pharynx and from the esophagus through the glossopharyngeal nerve and more fibers that come from its upper laryngeal branch [42]. As will be seen with regard to the effect of radiation treatment, these connections are important for obesity in pathophysiological terms.

### 3.4. The Hypothalamus

The various nuclei of the hypothalamus (not described in detail for brevity) play an important role in maintaining energy homeostasis by responding to peripheral hormonal and neural signals, and are activated to control any reduction in adiposity [9]. According to the set point theory, the hypothalamus keeps the quantity of adipose tissue (energy reserve) constant by controlling its utilization and accumulation [9]. This is a protective mechanism of evolutionary origin that has protected the human species from extinction due to famine and wars [9]. According to this theory, in obesity, the set point is fixed at a higher level, and this represents another significant obstacle to effective and long-term weight loss [12]. In the mediobasal hypothalamus, adjacent to the third ventricle, there is the arcuate nucleus (ARC) consisting of two populations of neurons: orexigenic neurons that produce the agouti-related neuropeptide (AgRP)/neuropeptide Y (NPY), and anorexic neurons that produce proopiomelanocortin (POMC) [50]. Both of these neurons receive nutritional information from the periphery: the signals of satiety stimulate POMC neurons and inhibit AgRP neurons [29].

These neurons oppose energy deficiency through the release of AgRP, NPY, and GABA to increase food search and consumption [50,51]. GIRK channels (G protein-gated inwardly rectifying K), are a family of K channels that mediate the inhibitory effect of neurotransmitters and hormones on cells of the nervous system [52]. The GIRK4 subtype, present in the ventrolateral and ventromedial part of the arched nucleus containing the agouti-regulated peptide (AgRP) and the neuropeptide Y (NPY), has a role in the regulation of energy homeostasis, because mice with a mutation of gene *GIRK4* show a predisposition to late-onset obesity [52,53]. The immunoreactivity of GIRK4 was also observed in neurons in the ventromedial part of the arcuate nucleus containing the agouti-regulated peptide (AgRP) and the neuropeptide Y (NPY) [53].

Furthermore, within the hypothalamic lateral, dorsomedial, and peri-fornical regions neurons are present that secrete the orexin neuropeptide, which causes an increase in spontaneous physical activity (SPA) and energy consumption [54]. Thus, orexin significantly limits weight gain and adiposity even when a high-fat diet (HFD) is consumed (which normally reduces physical activity) [54,55]. Furthermore, this reduces NEAT (non-exercise activity thermogenesis), so spontaneous physical activity (SPA) burns fewer calories [54]. In addition, through the orexin of hypothalamic origin, the vagal nuclei of the brainstem perceive changes in blood glucose and modulate pancreatic innervation by activating neurons in the DMV that project onto the endocrine pancreas [42].

### 3.5. The Cerebellum

The medial nucleus, also called the fastigial nucleus (FN) of the cerebellum, through bidirectional connections with the hypothalamus (in particular its ventro-medial nucleus) and with the reward system, is involved in the control of food intake, energy homeostasis, and body weight regulation [56,57]. This has been clearly demonstrated in functional imaging studies [58]. Satiety induced by a liquid meal is associated with a decrease in blood flow in the cerebellum, which was found to be significantly greater in obese than in lean subjects ($p < 0.005$) [59]. Its dysfunction enters into the pathogenesis of obesity [33,60].

### 3.6. Areas of the Cortex

The left dorsolateral prefrontal cortex (DLPFC), which is part of the dorsal cognitive fronto-striatal circuit, exerts cognitive control over food intake [38]. Obesity is associated with a reduced activation of this brain region [61]. Its postprandial activation is significantly inferior in the obese compared to in the lean [62]. A lack of cortical cognitive control has also been verified in other eating disorders such as bulimia nervosa (BN) and binge eating disorder (BED) [38].

### 3.7. Functional Brain Differences in Patients with Obesity as Evidenced by Neuroimaging

Most fMRI studies evaluating appetitive (i.e., craving) or anticipatory responses to visual food stimuli have shown that the brain of normal-weight subjects compared to that of subjects with obesity reacts differently in the regions associated with reward (insula, striatum, orbitofrontal cortex), executive control (cingulate cortex, prefrontal cortex), and energy homeostasis (hypothalamus) [63]. Subjects with obesity have a greater activation of the reward system in response to images of high calorie foods [9,64].

Additionally, people with obesity exhibit brain activity that is similar to drug addicts and gambling addicts [36,65]. For example, they show a reduction in the activation of the striatum when a highly palatable food is consumed, which is a result surprisingly similar to the effects of drugs and gambling in addicts [36,65].

Positron emission tomography (PET) shows that obesity is associated with abnormal neuronal activity in certain brain regions related to food intake [66]. For example, the response of the hypothalamus to a meal is much slower in subjects with obesity [9]. In the prefrontal cortex and in some limbic/para-limbic areas of the brain, the changes in blood flow in response to a meal also differ between individuals with obesity and normal-weight individuals. In addition, PET, using D2 or D3 receptor ligands, shows a reduced presence of dopamine receptors in the striatum of subjects with obesity that correlates negatively with the body mass index [9].

### 3.8. Prader–Willi Syndrome (PWS) as an Additional Source of Information

Prader–Willi syndrome is a genetic disease, linked to a CNS dysfunction, which causes an uncontrolled intake of high-calorie food and leads to considerable overweight (subjects reach more than 200% of their ideal weight) [67]. The defect is borne by chromosome 15, where the genes of GABRB3 and GABRA5 subunits of the GABA neurotransmitter (involved in food intake and energy homeostasis) are expressed abnormally [67,68]. In most cases, it is a 15q11-q13 deletion in the long arm and, in a minority of cases, a uniparental disomy (a chromosome pair that derives from a single parent, in this case from the mother) [67,68]. It is caused by the absence of the paternal copy of the genes due to partial deletion of the long arm of the chromosome which leads to loss of gene expression and to epigenetic alterations [69].

Studies in human and mouse models suggested that the PWS hyperphagia derives from a dysregulation of energy homeostasis in the hypothalamus [69]. Weight changes precede appetite changes, which implies that the development of hyperphagic behavior is preceded by energy control changes [69]. Through neuroimaging, an increase in reward network activation and

a deficit in cognitive control of food intake has been revealed [70,71]. However, the main neural mechanisms implicated in PWS overeating remain unclear [67].

### 3.9. Another Rare Form of Childhood Obesity Related to Genetic Alterations Involving the CNS

Another type of congenital obesity involving the CNS (dorsomedial nucleus of the hypothalamus), caused by a significant increase in appetite, depends on an alteration of the leptin-melanocortin pathway [72]. The genetic alteration prevents expression of the stimulatory G protein-subunit (Gsα), which causes the reduction in Gsα-dependent signaling at the level of the melanocortin 4 receptor (MC4R) [72].

## 4. Effectiveness of Metabolic Surgery: Which Could Be the Mechanisms?

It is well known that weight loss and glycemic control are difficult to achieve by obese patients, and if achieved, are difficult to maintain in the long term, even by drastic lifestyle intervention. This is in contrast with bariatric surgery, which is currently the most effective therapeutic approach for obesity (consistent and lasting weight loss) and Type 2 DM (average improvement in HbA1c was 2.1% in operated patients compared to 0.3% in patients undergoing intensive medical care) [73]. Metabolic improvements occur within days or weeks following surgery, even before significant weight loss occurs. Therefore, mechanisms are triggered that go beyond simple weight loss and calorie restriction [73].

This prompts the question of what are the mechanisms underlying the effectiveness of metabolic surgery. There is undoubtedly a concurrence of various effects. A common base appears to be the fact that surgical procedures sever the branches of the vagus nerve, altering its connection with the brainstem's dorsal vagal complex (DVC) and other brain areas with which it is linked, in particular those of the reward system [74]. Some consequences of these effects are described in the following sections.

### 4.1. Changes to Feeding

Several studies have provided convincing indications that metabolic surgery leads to an altered gut–brain communication that modifies the reward mechanism, among others [75]. One of the most important effects is the change in tastes, and the apparent aversion to more caloric foods (especially lipids) and the shift towards healthier foods such as fruit and vegetables, in addition to a reduction in the duration and size of meals (even in the presence of weight loss which is usually a powerful stimulus to eat) [9,33,76].

fMRI studies have confirmed that Roux en-Y Gastric bypass (RYGB) decreases the activation of brain regions involved in hedonic control, including the lentiform nucleus, the putamen, and the dorsolateral prefrontal cortex, in response to high-calorie food stimuli [77].

Why are these changes taking place? One of the hypothetic mechanisms is that the anatomical reconfiguration of the GI tract produced by surgery corrects the dopaminergic dysregulation in the reward system and restores its sensitivity [33,78]. As already described, the reward circuit plays a particular role in the development of food preferences and in the drive towards the consumption of palatable foods. Gastric bypass patients demonstrate increased availability of dopamine D2 receptors after weight loss, indicating that the effects of overeating on dopamine receptor downregulation may be reversible [31]. Thus, at the time when a balance is re-created, the peripheral signals are again able to cause satiety and overeating stops [35].

### 4.2. GLP-1: Modifications Following Bariatric Surgery in Light of Its Role in Obesity

The entero-hormone glucagon-like peptide (GLP-1) provides an important control of energy and glycemic homeostasis. Among its various actions, that on the pancreas consists of the increase in insulin secretion by the pancreatic beta cells in a glucose-dependent manner (incretin effect) and in the inhibition of hepatic glucose production through a reduced secretion of glucagon by the alpha cells,

which is particularly important in the postprandial phase. Furthermore, as already mentioned, once secreted by L cells, GLP-1 activates the receptors on the vagal fibers present in the intestinal epithelium. As previously described, this affects food intake via the NTS and the other brain centers. Finally, the small amount of the hormone (about 10%) that enters the circulation has a direct effect on the hypothalamus, causing a reduction in appetite. The same effect (plus the prevention of the rapid entry of glucose into the systemic circulation) is obtained by slowing gastric emptying [79].

A reduced incretin effect has been observed in obese–diabetic patients with reduced levels of GLP-1, but it is not clear whether this plays a pathophysiological role or is only the consequence of the increase in adipose tissue [79]. Consequently, both drugs, which inhibit the enzyme dipeptidyl-peptidase 4 (DPP4) (responsible for degrading it within a few seconds) and the synthetic analogs (which, not being degraded, remain in circulation longer), have been successful in glycemic control and weight loss [80]. Among the latter, 3.0 mg of liraglutide once daily (added to a reduced-calorie diet and increased physical activity) leads to significant weight loss associated with improved diabetes [81]. Another analog, semaglutide (which has the advantage of being administered only once a week) has been shown to be particularly effective in achieving clinically relevant weight loss compared to placebo [82].

GLP-1 has been evaluated after various bariatric surgeries to determine if it played a role in the dramatic energy-metabolic improvement. It was found that GLP-1 levels increase up to six times after RYGB surgery [64]. Although the topic has been the subject of much debate [83], despite this high increase, GLP-1 cannot be considered the main actor of the improvements because the same effects are obtained in animals deficient for the GLP-1 receptor [84]; the administration of the analog of GLP1, liraglutide, does not achieve the same effects [84] and when Exenedine-9 (the GLP-1 receptor antagonist) was administered after RYGB, although worsening of glycemic control was observed, there was no return to the initial altered glucose tolerance [85]. It should also be borne in mind that the activity of enteric hormones is predominantly local (paracrine, on vagal fiber receptors) as they are rapidly degraded. Only a limited portion of these hormones also enters the circulation, carrying out their action remotely.

## 4.3. Reactivation of the Vagus Afferent Neurocircuits Disrupted by HFD

As mentioned above, chronic consumption of a HFD causes alterations in the properties of vagal neurons. There is evidence that HFD consumption leads to alterations in both vagal nerve function and structural integrity; in particular, the membrane input resistance, excitability, and reactivity to the satiety neuropeptides leptin, CCK, and GLP-1 of vagal afferent neurons are reduced, in addition to the postprandial neuronal activation of NTS [86]. HFD also induces leptin resistance which impairs responsiveness of mucosal vagal terminals to meal-related gastrointestinal signals [87]. Furthermore, a reduced sensitivity to satiety peptides, such as CCK and intestinal nutrients, has been proven as well as a reduced neuronal postprandial activation of NTS [88]. These alterations are sufficient to cause hyperphagia and weight gain, a demonstration that abnormal gut–brain signaling is the triggering factor for obesity [88].

These abnormalities regress after RYGB with positive effects on the visceral and gastrointestinal reflexes mediated by the vagus [86]. Furthermore, this RYGB efficacy suggests that vagus-vagal neurocircuits remain open to readaptation and that the effects of obesity do not necessarily entail permanent and unrecoverable impairment of the brainstem nerve circuits [86]. In addition, the ready reversibility of the alterations suggests that vagal neurocircuits may represent valid and easily accessible targets for obesity research [86].

## 4.4. Increased Insulin Response and Sensitivity

The acute insulin response (an index of beta cell sensitivity) increases after bariatric surgery [89]. In fact, after the oral glucose tolerance and mixed meal tests, an earlier and more amplified increase in insulin concentration than that seen in the pre-operative phase can be observed [89]. Although the

relative mechanism is not yet well understood, it could be due to the restored vagal modulation of the endocrine pancreas [90].

*4.5. Reduced Hepatic Glucose Production (HGP)*

The precocious improvement in glycemic control is also caused by the increase in liver sensitivity to insulin and the consequently reduced HGP [89], which is under duodenal control (Figure 3).

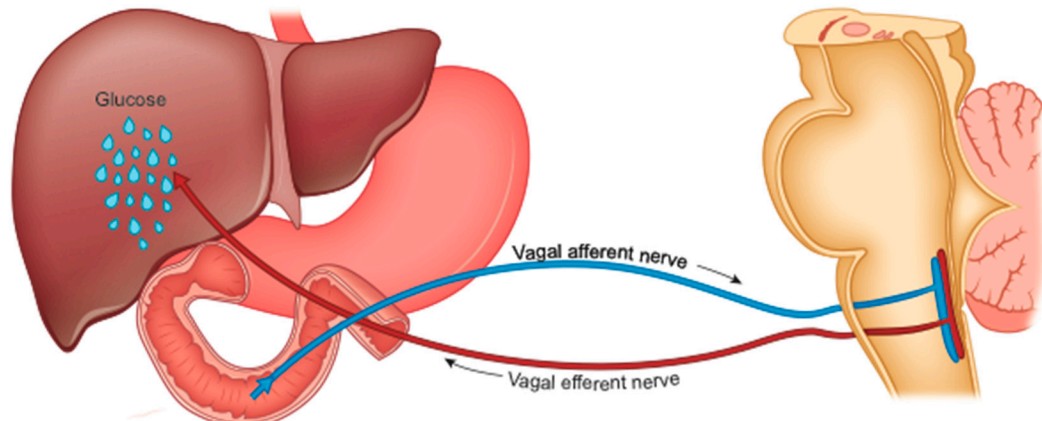

**Figure 3.** Nervous control of hepatic glucose production by nutrients in the first part of the intestine, through the brainstem vagal nuclei.

It is now certain that the exclusion of the foregut is the main culprit of the benefits of metabolic surgery [91,92]. The various interventional practices directly or indirectly affect the duodenum. After RYGB, the therapeutic effects are canceled when food is administered directly into the bypassed segment of the intestine [93].

The duodenum plays a fundamental role in maintaining glycemic homeostasis by controlling HGP [94]. It is the tract of the intestine most innervated by the vagus. The passage of food produces inputs towards the NTS neurons which retransmit them to the liver to reduce HGP [95]. This mechanism is interrupted in rodents with obesity and with diabetes, suggesting an alteration in intestinal sensing or in the transmission of signals from the intestine to the brain [96,97].

Diabetes and obesity are characterized by excessive hepatic glucose production, especially postprandial (fed stage). Many observations suggest that at the origin of this increase in HGP there is also dysfunction of the gut-brain-liver axis [96]. Metabolic surgery, which interrupts (or reduces) the contact between food and the duodenal mucosa, could interfere with HGP [95] and determine the rapid improvement of the metabolic context in operated patients (regardless of weight loss) [91]. This mechanism is similar to that activated by weight loss resulting from a drastically low-calorie diet [98]. However, for a given weight loss, surgery is twice as effective [99].

4.5.1. HGP and Cholecystokinin

The presence of lipids in the duodenum, stimulating the release of cholecystokinin (CCK) from I cells, inhibits HGP through the gut-brain-liver axis [95,100,101].

Rats with obesity are resistant to this effect of fats [90,91]. Duodenal bypass surgeries could also act by eliminating this resistance [101]. Proteins act through the same mechanism.

4.5.2. HGP and TOR Enzyme

The enzyme "target of rapamycin" (TOR) is a serine-threonine kinase active in the duodenum of rodents with obesity that induces the increase in HGP through the gut-brain-liver axis [102]. TOR inhibition reduces hepatic glucose production and restores glycemic normality in these rodents [102]. Confirming this effect of the enzyme, the metabolic syndrome represents a common

side effect of various therapeutic uses of TOR inhibitors such as in immune-inhibition in transplants, anti-neoplastic therapy, treatment of neurodegenerative diseases, etc. [103].

### 4.5.3. HGP and Serotonin

Serotonin (or 5-hydroxytryptamine, 5-HT), produced by the chromaffin cells of the proximal small intestine, plays an important role in the regulation of HGP through the activation of vagal afferent fibers to the brainstem (which express the related receptors) [104]. Obesity is characterized by an increased production of 5-HT in the small intestine, which is proportionate to BMI and level of hyperglycemia [104]. In rodents, the pharmacological block of the tryptophan-hydroxylase enzyme that catalyzes the formation of 5HT protects against obesity and hyperglycemia induced by HFD [104].

### 4.5.4. Accelerated Gastric Emptying in Sleeve Gastrectomy (SG)

In SG, which is currently the most practiced type of metabolic surgery (in which the stomach is reduced to about 20% by dissecting it along the greater curvature), the passage of food into the duodenum is preserved but, notwithstanding, it exerts an anti-diabetic effect similar to RYGB which is not due to the restriction of the stomach volume [105].

The integrity of the entire gastric wall is essential to achieve an optimal level of digestion before the passage of food into the duodenum [42]. In turn, functional relationships exist between the duodenum and the stomach to optimize the digestion and absorption of nutrients in the small intestine [42]. The anatomical upheaval of the gastric structure by SG, modifying the food gastric processing, inevitably affects the neuro-hormonal functions of the duodenum.

It can be hypothesized that SG is effective on diabetes because of the rapidity of transit of the food bolus in the duodenum that interferes with the normal stimulation of the enteric-endocrine cells, which, through the vagus, regulate HGP. In fact, it has been proven that the speed of gastric emptying is a fundamental factor for the regulation of postprandial glucose as it regulates the metabolism of carbohydrates in the liver [106].

Furthermore, SG, by removing the gastric fundus where the peptide hormone ghrelin is secreted, could affect postprandial blood sugar because this hormone activates the vagal gut-brain-liver pathway [29], regulating HGP [97]. A recent meta-analysis has shown that the level of ghrelin decreases significantly after SG [107].

### 4.5.5. Conclusive Hypothesis on the Role of HGP

In conclusion, the possibility that the reduction in HGP caused by surgical interventions involving the proximal part of the small intestine may play an important role in the reduction in diabetes, as described, appears convincing. This hypothesis should be confirmed by an in-depth study of the involved molecules, to develop drugs that achieve the same goal non-surgically [100,101].

### 4.6. Modification of the Hypothalamic Set Point

Following bariatric surgery and consequent changes in vagal inputs, the set point could be reset to a normal level, and the defense of excess body fat interrupted [12,75]. Patients report experiencing less (rather than more) appetite, even in the face of significant weight loss [12]. Furthermore, a resumption of spontaneous physical activity cannot be excluded through the reactivation of the hypothalamic neurons that produce orexin, thus contributing significantly to weight loss [108].

### 4.7. Cerebellum

Several results suggest that RYGB also affects specific areas of the cerebellum (right and median cerebellum, lobe 8) involved in reward-related behaviors, which may partially explain the changes in eating behavior following surgical intervention [33].

### 4.8. Alteration of Intestinal Bacteria

The gut microbial ecosystem contains 10 times the number of cells in the body and about 150 times more genes than the human genome. It plays an important role in maintaining a state of metabolic homeostasis and in monitoring body weight, principally affecting the gut-brain axis. Furthermore, it is able to influence the activation of the vagal afferent nerve endings present in the lamina propria [88] by stimulating the secretion of entero-hormones, first of all serotonin and GLP-1. In addition, intestinal bacteria produce enzymes that synthesize quinolinic (an NMDA receptor agonist) and kinurenic (antagonist) acids.

The microbiome (the microbial community occupying the gut) in obese and type 2 diabetic individuals is structurally and functionally different from that of normal subjects. Its genetic richness is lower [109]. After bariatric surgery, the variety of bacteria improves and a change in the enterotype is observed [109] but most patients remain with very low microbial gene richness (a measure of genetic diversity), despite the important clinical improvements [109]. The colonization of microbiota-free animals with microbiota from obese donors leads to excessive weight gain; conversely, intestinal microbiota transplantation from subjects treated with RYGB to unoperated subjects results in weight loss [89], strongly suggesting that intestinal dysbiosis (the interruption of the physiological symbiotic relationship between the human host and the microbiota) plays an important role in the pathogenesis of obesity [110].

The exact mechanisms remain uncertain. One of the effects of the high-fat obesogenic HFD is to significantly reduce bacterial diversity in the gastrointestinal tract. *Firmicutes* (bacteria that have a greater ability to collect energy from the diet and to generate more storable energy by digesting commonly poorly digested polysaccharides) are increased compared to *Bacteriodetes* (both represent over 90% of intestinal microbes) [111]. The pro-inflammatory microbiome proliferates, altering the vagal signaling and affects the gut–brain communication [112].

### 4.9. Possible Effect of Neck Radiation Therapy

A reported peculiar effect of neck radiation (in the case of neoplasia) is the regression of diabetes [113–115]. The mechanism might be very similar to that previously suggested for the GI after bariatric surgery: a deficit of vagal afferent impulses from the oropharynx neurosensory cells to the NTS and consequent return of the abdominal organs to their proper functions, in particular hepatic glucose production [47]. Given the abundance of afferent vagal fibers of oropharyngeal origin that reach the NTS, influencing its activity (as illustrated before), it is plausible that the action of ionizing radiation on these nerve terminals may have an effect on metabolic control.

## 5. What Could Be the Main Targets of Non-Conventional and Non-Surgical Therapeutic Approaches to Obesity?

According to the declaration of international diabetes organizations regarding metabolic surgery (2nd Diabetes Surgery Summit DSS-II, an international consensus conference), an important goal of the research is to increase understanding of diabetes therapeutic mechanisms to develop alternative treatments [116]. In fact, surgical treatment is often not practicable and at risk of complications, although limited [83]. For instance, operated patients have a higher risk of developing nutrient deficiency-related disorders such as anemia, certain types of neuropathies, or osteoporosis [73].

Objectives of pharmacotherapy of obesity include central and peripheral regulation of food intake, energy consumption, and physical activity [50].

A brief overview of the possible targets of different drugs is given below (Table 1).

**Table 1.** Potential sites of attack for pharmaceutical therapy of obesity (for details and references please see text).

| Main Sector | 1st Subgroup | 2nd Subgroup |
| --- | --- | --- |
| Post-synaptic glutamate and GABA receptors | NMDA receptors | Subunits GluN2A and GluN2B |
| | NMDA receptor co-agonists | Inhibition of glycine transporters or block of the glycine site |
| | | Memantine: blocks excessive activation of the NMDA receptor by glutamate |
| | Gamma-amino-butyric acid (GABA) receptors | GABAB's positive allosteric modulators (PAM ADX71441) |
| 5-HT3 agonists and antagonists | 5-HT2C agonist lorcaserin | Type 3 serotonin receptor antagonist Ondansetron |
| Guanine protein-coupled receptors (GPCR) | Activation of designer drug receptors (DREADD) | Muscarinic M3 DREADD receptor (hM4Di) |
| Glycine transporter 1 (GlyT1) | Inhibitor of GlyT1 | |
| Melanocortin 4 receptor (MC4R) | MC3/4R agonists | |
| Opioid system | Opioid antagonist, Naltrexone μ receptor agonists | |
| Neuropeptide orexin | DREADD-dependent activation of orexin neurons | |
| GIRK4-containing channels | | |

## *5.1. The Synaptic Receptors*

A pharmacological intervention to modulate neurotransmission through the synaptic receptors in the involved brain sectors could correct the maladaptive plasticity at the origin of food addiction, restoring normal sensitivity and correct functioning of the system. In particular, this fits synaptic regulation of vagal brainstem neurocircuits regulating autonomic functions that appear to be malfunctioning in different pathophysiological conditions, including obesity [42].

### 5.1.1. The N-Methyl-D-Aspartate Receptor (NMDA)

As illustrated, the post-synaptic N-methyl-D-aspartate (NMDAR) receptor, activated by the neurotransmitter glutamate, plays critical roles in the physiological function of the mammalian CNS, including weight control [117] and synaptic plasticity, through modulating excitatory neurotransmission. Among the receptor's various subunit identities (which confer distinct pharmacological and biophysical properties on the receptor and, consequently, on neuronal processes [117,118]) the subunits GluN2A and GluN2B are involved in the control of nutrition and energy homeostasis [117,119]. They are the most common in the adult brain, and their expression varies in different sectors [119]. GluN2B is the most well-investigated subtype in studies on NMDARs. The subunits GluN2A and GluN2B are also present in the neurons producing agouti-related peptide (AgRP), thus representing possible therapeutic targets [117].

In addition, the NMDAR is responsible for addictive behaviors [40], including food addiction (comprising the "binge" type), as a result of the plasticity of the two subunits, in particular the NR2B [30]. Therefore, selective antagonists of this subunit (including ifenprodil and Ro 04-5595) can be primary therapeutic tools [30,120]. The study of NMDARs, and specifically of GluN2 subunits within critical areas of brain neurons, is important for our full understanding of the normal regulation of energy balance and can potentially lead to insights into new anti-obesity pharmacological targets [117]. For example, a number of positive allosteric modulators (PAMs) (a group of substances that connect to

binding sites different from those of the other agonists of the receptor to modify its response to the stimulus) have been identified. These could act on the dysfunctions of the various GluN2 subunits of NMDARs present within different areas of the central nervous system, to treat a wide range of non-neurological disorders, including diabetes but avoiding its toxicity [121,122].

Another potentially effective therapeutic tool, memantine, is a derivative of adamantane and is already used in Alzheimer's disease, blocking excessive activation of the NMDA receptor by glutamate, which is the process that occurs in addiction [30]. Its action as a non-competitive antagonist of extra-synaptic NMDAR, does not interfere with normal synaptic transmission [30]. In obese mice, it reduces weight by suppressing food intake and increasing physical activity. Hence, it could significantly correct over-eating in human and animal models [30].

However, the use of active drugs on the NMDAR is made difficult by the fact that the majority of CNS structures require these receptors to function properly.

Hence, nonspecific NMDAR inhibitors can have a number of negative side effects, including memory deficits, concentration difficulties, agitation, and catatonia, in addition to psychotomimetic effects such as hallucinations [30]. This limitation could be overcome by identifying highly specific drugs for the subunits involved in the control of body weight, hopefully targeting only those neural circuits responsible for pathological disruption of energy homeostasis with minimal effects on general receptor sensitivity [123]. To date, despite strong preclinical support for the beneficial effects of glutamatergic NMDA receptor ligands, no effective results have been obtained in clinical trials [40].

### 5.1.2. NMDA Receptor Co-Agonists

NMDA receptor activity requires the presence of d-serine (or d-glycine) as a co-agonist at the glycine binding site [30]. Therefore, changes in the regulation of these cofactors can alter the function of the NMDA receptor, influence addiction-related behaviors, and be subject to pharmacological intervention (for example, by inhibiting a glycine transporter or by blocking the glycine site [30]).

### 5.1.3. Gaba-Amino-Butyric Acid (GABA) Receptors

GABA is the main inhibitory neurotransmitter in the central nervous system and its malfunction can cause alterations in energy homeostasis. Pharmacological activation of the GABAB receptor with baclofen (β-[chlorophenyl]-GABA) or other GABAB-agonists, can counteract various forms of addiction ([124] AGABIO). However, even in this case, various side effects occur including sedation, respiratory depression, and motor deficit, which hinder their therapeutic use. Nonetheless, GABAB's positive allosteric modulators (PAM), such as PAM ADX71441, represent a valid therapeutic option [124].

### *5.2. Other Potential Targets of Drug Treatment Beyond the Glutamatergic Signaling Circuits*

### 5.2.1. Dopamine Receptors

As illustrated previously, dopamine and its receptors play a leading role in controlling food intake and related disorders. It is therefore logical to think that these may be the goal of a therapeutic approach to obesity. The research has made many attempts but the results are still at an early stage. Regarding the ligands of the D1 and D2 receptors (agonists, antagonists, and partial agonists), selective compounds with new pharmacological targets in relation to the interactions with the dopamine-glutamate systems have been experimented in different diseases; this may lead to the discovery of a drug suitable for addressing the complexity of dopamine receptors' hetero-complexes in native systems using multiple intra-cell markers and benefiting from more selective tools available for studying the dopamine receptors [125]. In particular, the DRD4 receptor, which is part of the D2-like family, has a predominant expression in the prefrontal cortex (PFC), a brain area closely involved in the modulation of reward processes related to both food and drug consumption [126]. Several human studies have revealed a possible involvement of DRD4 in eating disorders and obesity [126]. Hence, DRD4 is an important

pharmacological target to be explored for innovative selective drugs in the treatment of addictive disturbances such as eating disorders and related comorbidities [126].

Finally, clinical studies on tesofensine, which inhibits the reuptake of dopamine (plus serotonin and noradrenaline), indicate significant efficacy in promoting weight loss mainly through a reduction in appetite [127]. However, side effects have been described: increased heart rate, nausea, insomnia, flatulence, and depression which portends that its use will be problematic [127].

### 5.2.2. 5-HT3 Agonists and Antagonists

Serotonin (5-hydroxytryptamine) 2C receptor agonists, such as MK-212 and fenfluramine, inhibit food intake by acting on POMC-expressing neurons in the arcuate nucleus. Nor-fenfluramine (derived from fenfluoramine) is responsible for its effective anorexiant activity [128]. Unfortunately, it was at the origin of a potentially lethal heart disease (thickening of heart valves through activation of the $5-HT_{2B}$ receptor in cardiac valvular interstitial cells), leading to its withdrawal.

The study of these substances led to the development of the anorexiant lorcaserin, a 5-HT2C agonist, which acts on POMC-expressing neurons with potent 5-HT2B and 5-HT2A agonist activities, expressed almost exclusively in CNS, which acts through hypothalamic activation of the anorexic pro-opiomelanocortin (POMC) pathway [128]. It is considered to be safe and has a modest, but lasting, weight loss activity in obese subjects and improves glycemic control; it is FDA-approved for treating obesity [129].

Ondansetron, a type 3 serotonin receptor antagonist, which inhibits vagal afferent activity, utilized in vagus-mediated emesis, can be prescribed for the treatment of overeating (in BN it re-normalizes the sense of satiety and reduces the size of meals) [130].

### 5.2.3. Guanine Protein-Coupled Receptors (GPCR)

G-proteins (nucleotide-binding proteins) act inside the cells as molecular switches; activated by a transmembrane receptor, they transmit signals from outside (such as from neurotransmitters). By modulating the activity of their receptors with a synthetic ligand (RASSL) or with drugs active exclusively on designer drug receptors (DREADD), control of their signaling is gained [131]. DREADDs are a form of pharmacosynthetic (or chemogenetic) long-acting neuromodulator [132]. They depend on the G-protein-coupled receptors that are modified to lose affinity for their biological ligand and obtain a powerful activation by a synthetic drug. This technique allows the neural circuits in control of eating behavior to be better identified and influenced [128]. For example, by acting through the human muscarinic M3 DREADD receptor (hM4Di) (which can silence neuronal activity) on AgRP neurons, food intake can be inhibited [128].

These experimental techniques can lead to identification of new anorexiant drugs [133] by affecting different GPCR signaling pathways in a variety of hypothalamic nuclei to treat obesity [128]. Experiments are currently underway, with positive results, in subhuman primates [128].

### 5.2.4. Inhibitor of Glycine Transporter 1 (Glyt1)

Experimentally, the administration of a glycine transporter 1 inhibitor (GlyT1) in the vagal nuclei of the brainstem suppresses HGP, increases glucose tolerance, and reduces both food intake and body weight in diabetic patients with obesity [134]. This drug is safely utilized in the treatment of schizophrenia [134].

### 5.2.5. The melanocortin 4 Receptor (MC4R)

A subpopulation of hypothalamic melanocortinergic neurons projects on the neurons of the NTS and on the dorsal motor nucleus of the vagus (DMV) [135], where melanocortin-4 receptors (MC4R) are expressed. Thus, melanocortinergic endings are positioned appropriately to influence the neural circuits involved in food intake [135] and MC4 receptor agonists are potential drugs for the treatment

of obesity [136]. The injection into the brainstem of an MC3/4 receptor agonist reduces the size of animals' meals for 24 h. In contrast, the injection of its antagonist increases meal size [135].

### 5.2.6. The Opioid System

In animals, injection into the bloodstream or into the cerebral ventricular spaces of opiate agonists (especially those which act on the µ receptors) increases the consumption of sweet and fatty foods, while the injection of antagonists reduces it [7]. The brain regions involved are: (1) the central nucleus of the amygdala (CeA) in which the opioid antagonist, Naltrexone, causes a reduction in the consumption of high-calorie food [7] and (2) the nucleus accumbens (NAc) in which, on the contrary, the µ receptor agonists cause an increase in its consumption [7,16] and its antagonists reduce it [16].

### 5.3. Activation of Physical Activity through Stimulation of the Neuropeptide Orexin

Because, as described, the reduction in spontaneous physical activity (SPA) contributes to obesity, the neurons that secrete orexin can represent a therapeutic target for its treatment, activating physical activity and increasing energy consumption [132]. For this reason, drugs have been studied which, by increasing the signaling levels of these neurons, could be effective. DREADD-dependent activation of orexin neurons could also achieve weight loss through an increase in SPA and energy expenditure, even without changing the amount of food consumed [132].

### 5.4. The GIRK Channels

Pharmacological approaches targeting GIRK4-containing channels should also be considered in the treatment of obesity [52].

## 6. The Effects of Physical Instruments (Table 2)

### 6.1. Electrical Stimulation of The Vagus Nerve

### 6.1.1. Blocking of Vagal Activity

The central role of the intestinal vagus nerve in the origin of obesity (as described above) makes it a valuable target of physical treatment [137]. Percutaneous CT-guided cry-ablation of the posterior vagal trunk causes appetite reduction and weight loss in subjects affected by mild or moderate obesity [138]. A similar effect is obtained by repeated electrical stimulation of the left cervical branch [139].

**Table 2.** Physical methodologies in the therapy of obesity (for details and references please see text).

| Main Sector | 1st Subgroup | 2nd Subgroup |
|---|---|---|
| Electrical stimulation of the vagus nerve | Blocking of vagal activity | Activation of the afferent vagus |
| Brain neuro-modulation | Deep brain stimulation (DBS) | Direct non-invasive stimulation (NIBS) with transcranial current |
| fMRI neurofeedback | | |

### 6.1.2. Activation of The Afferent Vagus

The vagus, following prolonged stimulation, activates a complex brain network. Vagus stimulation is of interest for the treatment of a variety of diseases including, in addition to obesity, epilepsy and severe depression [42].

The ability to selectively modify the plasticity of various brain sectors through afferent vagus stimulation (VNS), which triggers the release of acetylcholine and norepinephrine, represents a remarkable opportunity to treat a variety of brain dysfunctions, including those at the origin of obesity [139]. For example, VNS, coupled with sensory, motor, and cognitive exercises, acting on the

reward network affected by maladaptive plasticity, can normalize reward function [140,141]. Research is already underway to apply this principle to the treatment of bulimia nervosa [130].

*6.2. Brain Neuro-Modulation*

Neuromodulation techniques that show great promise are deep brain stimulation (DBS), repetitive transcranial magnetic stimulation (rTMS), transcranial direct electrical stimulation (tDCS), and neurofeedback [142]. These represent experiments that could soon be clinically used in the treatment of obesity due to the growing understanding of the mechanisms involved [142].

6.2.1. Deep Brain Stimulation (DBS)

DBS is an invasive but non-damaging neurosurgical procedure which, through electrodes implanted in the brain, sends electrical impulses that reversibly modify the activity of nerve cells and is capable of positively influencing eating behavior in obesity [142,143]. The electrical stimulation of NAc in diet-induced rats with obesity leads to marked reductions in food consumption and weight, an effect associated with an increase in DA levels and an upregulation of its D2 receptors [144,145]. DBS of the NAc in a patient suffering from hypothalamic obesity following a neurosurgical intervention for craniopharyngioma led to a decrease in appetite and a weight loss of 13 kg in 14 months without side effects [146].

6.2.2. Direct Non-Invasive Stimulation (NIBS) with Transcranial Current

Non-invasive brain stimulation (NIBS) (or transcranial magnetic stimulation (TMS)) is a technique in which the electric current of a coil placed on the skull generates a magnetic field that induces a parallel intracranial current [38]. Activation of the cortex is provoked, obtaining an improvement of various forms of addiction [147] and obesity [148]—understood as a food addiction condition—thus reducing food intake [149]. The main target of this treatment is the dorsolateral prefrontal cortex, responsible for self-control in food intake [38,150]. Beneficial effects have been obtained in various alterations of eating behavior, including binge eating [38]. Some limitations still need to be resolved, such as the need for repeated applications and the difficulty in accurately detecting targets, but this could represent a promising therapeutic path [38,149].

*6.3. fMRI Neurofeedback*

fMRI neurofeedback is a procedure that trains people in regulating the activity of a certain brain area in response to real-time feedback provided by functional neuroimaging (fMRI) using a brain-computer interface [142]. When used in experiments involving subjects with obesity, the choice of less caloric foods was obtained by improving the connection between the various areas associated with appetite and reward control [142].

**7. Treatment of Epigenetic Modifications**

Pharmaceutical-epigenetics is an area of research that refers to the study of chemotherapeutic agents that can reverse the epigenome alterations [151]. Since a crucial characteristic of epigenetic modification is that it is reversible and modifiable, the opportunity exists for preventive and therapeutic interventions on obesity and type 2 diabetes using specific drugs, as already effectively applied to other pathologies [152].

Initiatives at the population level to reduce the conditions linked to obesity, in the pre-conception phase, regardless of pregnancy planning, and focusing on diet and physical activity, appear essential to prevent intergenerational transmission of the phenotypic traits of obesity and diabetes by influencing epigenetic changes [153].

## 8. Conclusions

Obesity has reached epidemic proportions. The mechanisms can be found in the dysfunction of various sectors of the CNS that are responsible for energy homeostasis and are conditioned by epigenetic alterations. At present, the most effective and long-lasting therapy for obesity and related diabetes is metabolic surgery (which, however, is not universally practicable and is not devoid of side effects). Researchers' efforts should be directed towards further clarification of the neuro-hormonal bases of obesity and of the therapeutic mechanisms of surgery, aiming to identify drugs that will be active on the receptors of the various neurotransmitters involved, and modulating the activity of the vagal nervous system on the control of energy homeostasis and gastrointestinal functions (mainly the gut-brain axis). Particular attention may be warranted for the GluN2 NMDA receptor subunits. The final purpose would be to achieve the same surgical effects by means of non-surgical methods [50].

**Funding:** This research received no external funding.

**Conflicts of Interest:** The author declares no conflict of interest.

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
