# Peer review of "Obesity and Related Type 2 Diabetes: A Failure of the Autonomic Nervous System Controlling Gastrointestinal Function?"

_gastrointestdisord, doi:10.3390/gidisord2040039_

Round 1

Reviewer 1 Reportthe Authors addressed the concerns: please accept for publication.

Reviewer 2 Report

I have no another comments or suggestions.

Reviewer 3 Report

accept

This manuscript is a resubmission of an earlier submission. The following is a list of the peer review reports and author responses from that submission.

Round 1

Reviewer 1 Report

This work is focused on the role of the nervous system in obesity and the effectiveness of metabolic surgery and pharmacological interventions to contrast it. The review is interesting and well written; however, I have some comments for the author.

In particular:

- The title of the review refers to a work on obesity and related type 2 diabetes. However, the linkage between obesity and diabetes isn’t very treated in this paper, that discussed only the effect of bariatric surgery and on glycemic control. A brief paragraph on the relationship between diabetes and obesity could strengthen the review. - Pag. 5, line 166-168: authors should add the citation to this sentence. - Pag. 6, line 234-235: “regional blood flow of the cerebellum after food intake also showed differences between subjects with and without obesity”. It would be interesting to enrich this point describing the differences between the two groups of patients.

Pag. 8, paragraph 4.2: in my opinion, author should add a brief description on the role of GLP-1 on obesity, even discussing the effectiveness of analogue of GLP-1 on the treatment of the disease. (O'Neil PM, Birkenfeld AL, McGowan B, Mosenzon O, Pedersen SD, Wharton S, Carson CG, Jepsen CH, Kabisch M, Wilding JPH. Efficacy and safety of semaglutide compared with liraglutide and placebo for weight loss in patients with obesity: a randomised, double-blind, placebo and active controlled, dose-ranging, phase 2 trial. Lancet. 2018 Aug 25;392(10148):637-649. doi: 10.1016/S0140-6736(18)31773-2. Epub 2018 Aug 16. PMID: 30122305; Kluger AY, McCullough PA. Semaglutide and GLP-1 analogues as weight-loss agents. Lancet. 2018 Aug25;392(10148):615-616. doi: 10.1016/S0140-6736(18)31826-9. Epub 2018 Aug 16. PMID: 30122306; Pi-Sunyer X, Astrup A, Fujioka K, Greenway F, Halpern A, Krempf M, Lau DC, le Roux CW, Violante Ortiz R, Jensen CB, Wilding JP; SCALE Obesity and Prediabetes NN8022-1839 Study Group. A Randomized, Controlled Trial of 3.0 mg of Liraglutide in Weight Management. N Engl J Med. 2015 Jul 2;373(1):11-22. doi: 10.1056/NEJMoa1411892. PMID: 26132939).

Reviewer 2 Report

The author of the article should increase the information about the intestinal microbiota and its role in the development of the pathological process and in the action of drugs.

Reviewer 3 Report

The manuscript by Claudio Blasientitled “Obesity and related type 2 diabetes: a failure of the autonomic nervous system controlling gastrointestinal function?” has been reviewed.

The manuscript aims to describe the current state of the role of autonomous nervous system in the development of type 2 Diabetes.

In general, the manuscript is easy to read, and the ideas are well presented. The research topic is of importance, nevertheless, the manuscript holds several flaws. The main one is that the manuscript is too broad and do not provide concise information about the proposed aim.

This reviewer has some suggestions.

  1. Introduction. Along the manuscript, there is a colloquial and imprecise language that it is not suitable for scientific publications.
  2. The author, claimed that “ whereas the original source states that: “In 2010, the three leading risk factors for global disease burden were high blood pressure (7·0% [95% uncertainty interval 6·2–7·7] of global DALYs), tobacco smoking including second-hand smoke (6·3% [5·5–7·0]), and household air pollution from solid fuels (4·3% [3·4–5·3]).” Clearly, the author has no intentions to provide misleading information, but he has to be very careful about how to properly cite articles.
  3. Introduction. The following lines are problematic, not only because the text is duplicated, but due to the generalizations that are imprecise and could be miss interpreted.

“ Humans have a spontaneous attraction for high-calorie 32 foods, ingested in virtually unlimited quantities, and a natural tendency to avoid physical activity 33 and by a high-calorie diet. Actually, humans have a spontaneous attraction for high-calorie foods, to 34 be ingested in virtually unlimited quantities, and a to a natural tendency to avoid physical activity 35 [3,4].

  1. Please avoid the expressions “see below” or “see later” along the manuscript.
  2. Please avoid the excessive use of quotation marks “” instead refer to the exact terms, therefore there will be no need to quotations marks.
  3. Please avoid expression such as “For example, the response of the hypothalamus to a meal is much more "lazy" in subjects with obesity”.